# Disassembling the Nature of Capsid: Biochemical, Genetic, and Imaging Approaches to Assess HIV-1 Capsid Functions

**DOI:** 10.3390/v13112237

**Published:** 2021-11-07

**Authors:** Zachary Ingram, Douglas K. Fischer, Zandrea Ambrose

**Affiliations:** 1Department of Microbiology and Molecular Genetics, School of Medicine, University of Pittsburgh, Pittsburgh, PA 15219, USA; ZAI5@pitt.edu (Z.I.); dkf12@pitt.edu (D.K.F.); 2Pittsburgh Center for HIV Protein Interactions, School of Medicine, University of Pittsburgh, Pittsburgh, PA 15260, USA

**Keywords:** HIV-1, capsid, trafficking, nuclear import, reverse transcription, uncoating, microscopy

## Abstract

The human immunodeficiency virus type 1 (HIV-1) capsid and its disassembly, or capsid uncoating, has remained an active area of study over the past several decades. Our understanding of the HIV-1 capsid as solely a protective shell has since shifted with discoveries linking it to other complex replication events. The interplay of the HIV-1 capsid with reverse transcription, nuclear import, and integration has led to an expansion of knowledge of capsid functionality. Coincident with advances in microscopy, cell, and biochemistry assays, several models of capsid disassembly have been proposed, in which it occurs in either the cytoplasmic, nuclear envelope, or nuclear regions of the cell. Here, we discuss how the understanding of the HIV-1 capsid has evolved and the key methods that made these discoveries possible.

## 1. Introduction

Human immunodeficiency virus type 1 (HIV-1) is a lentivirus in the retrovirus family that causes chronic infection, leading to the gradual depletion of CD4^+^ T cells and acquired immunodeficiency syndrome (AIDS) if untreated. In 2020, approximately 38 million people were estimated to be infected with HIV-1, and nearly 700,000 AIDS-related deaths occurred [1]. Antiretroviral therapy (ART) has advanced greatly, improving morbidity and mortality outcomes. While ART is effective, universal availability has not been achieved, drug resistance leads to lower efficacy, and it does not provide a functional cure for HIV-1. As such, advancing our understanding of HIV-1 replication should continue, with the goal of developing novel therapeutics. The hallmarks of retroviral infection are reverse transcription of the viral single-stranded RNA (ssRNA) genome into double-stranded DNA (dsDNA) and the integration of viral dsDNA into host chromatin. To complete integration, the pre-integration complex (PIC), consisting of viral integrase (IN) bound to the viral genome, must enter the nucleus. Unlike some other retroviruses, HIV-1 does not require mitosis to access the chromatin, allowing the infection of nondividing cells. Early post-entry HIV-1 replication events through integration remain a primary focus for antiretroviral development, with research often centered on understanding these processes. Early replication events that pertain to the capsid—reverse transcription, nuclear import, and integration—have undergone extensive study in the past decade. With advancements in structural, biochemical, and microscopic techniques, our understanding of early HIV-1 replication has vastly improved.

## 2. Properties of Capsid

### 2.1. Overview of HIV-1 Capsid Functions during Early Replication Events

Following assembly and release from the plasma membrane of a producer cell, immature HIV-1 particles undergo proteolytic cleavage of several viral proteins. Cleavage of the Gag polyprotein leads to the separation of three structural proteins required to form an infectious, mature virion: the matrix (MA), capsid (CA), and nucleocapsid (NC) [2]. Roughly 1200 CA monomers form into 216 hexamers and 12 pentamers to produce an enclosed, conical capsid lattice containing two copies of the HIV-1 ssRNA genome [3,4,5]. The unique capsid structure is formed by the positioning of the pentamers at the closed ends, leading to a narrow end and a wide end, which deviates from classical helical or icosahedral capsid morphologies of other viruses. Mature HIV-1 capsids are known to be conformationally heterogeneous and dynamic, likely allowing its diverse functions during virus replication and the ability to interact with multiple viral and host factors [6].

After HIV-1 fusion with the cell membrane, capsids are exposed to a multitude of host factors in the cell cytoplasm, which can enhance or restrict infection of the cell [7]. To deliver the HIV-1 genome to the nucleus, active transport of the capsid occurs via interactions with microtubule motor proteins and adapters [8]. Traversal through the nuclear pore complex (NPC) into the nucleus requires CA interactions with NPC components and karyopherin proteins [9,10,11,12,13]. In the nucleus, the viral dsDNA is stably integrated into host DNA. During transit from the cell periphery into the nucleus, the capsid undergoes a set of poorly understood disassembly events, termed uncoating. The spatial and temporal aspects of capsid uncoating are crucial for completing reverse transcription, nuclear import, and integration. As such, various capsid uncoating models have been proposed, with evidence for cytoplasmic, nuclear pore, and nuclear uncoating [14]. This complex set of events determines replication success and has undergone extensive study, including the use of CA mutants, biochemical assays, and imaging techniques.

Beyond beneficial host interactions, the HIV-1 capsid must also contend with host restriction factors. By maintaining the capsid lattice, the viral genome remains protected from innate immune responses [15,16,17]. Host restriction factors such as tripartite motif proteins 5α and 34 (TRIM5α and TRIM34) and myxovirus resistance 2 (MX2) recognize features of the capsid lattice and can destabilize this protective shell [10,11,12,13,18]. Loss of the capsid lattice precedes an innate immune response to pathogen-associated molecular patterns, such as viral RNA. Retinoic acid-inducible gene I (RIG-I) and melanoma differentiation-associated gene 5 (MDA5) recognize HIV-1 RNA, albeit poorly due to the protection by the capsid and similarity to host RNA structure (reviewed here [19]). Cytoplasmic HIV-1 DNA can be recognized as non-chromosomal DNA, leading to an interferon response through cyclic GMP-AMP synthase (cGAS) and stimulator of interferon genes (STING) [16,20,21]. While host restriction factors may reduce replication, HIV-1 is not completely inhibited by any currently identified host factors.

### 2.2. Assessing Capsid Stability Using Biochemical, Genetic, and Imaging Approaches

Mutations in the HIV-1 capsid result in pleomorphic effects. While host interactions and immune evasion exert selective pressures on the capsid, structural constraints limit tolerable mutations for capsid integrity. Systematic substitution of CA residues often results in replication-defective virus [22,23]. Critical capsid surfaces have been mapped to helix 4 and 6 of the *N*-terminal CA domain, helix 8 of the C-terminal domain, and the dimer interface formed by adjacent CA molecules. Alteration of these key surfaces can inhibit capsid maturation, such as the CA K158A mutation in the major homology region [22]. Alternatively, capsid stability or host factor binding sites may be altered, leading to reduced infectivity, as is the case with E45A or N74D [22,24]. High levels of sequence conservation are also found among lab-adapted strains and clinical isolates. However, clinical isolates may undergo more variable selective pressures based on host protein polymorphisms, immune responses, and antiviral therapies. Thus, high sequence conservation suggests that the capsid is genetically fragile and constrained by the structural role of the capsid [25]. CA mutants have contributed to the understanding of HIV-1 replication by altering its phenotype and allowing the discovery of novel host interactions and cell-specific differences.

Assessing the stability and integrity of the HIV-1 capsid has advanced over the years with several biochemical and imaging approaches (Table 1 and Table 2). Early work in isolating capsids relied solely on ultracentrifugation of HIV-infected cellular extracts [26]. This method allowed the isolation of PICs and capsids, which provided a readout of intact capsids (Table 1) [25]. Later assays expanded on this method by comparing cytoplasmic and nuclear fractions for intact capsids in the fate-of-capsid assay (Table 1) [18]. The ability of HIV-1 CA or CA-NC to assemble into tubes or capsid-like structures in high salt concentrations in vitro allowed structural studies of the lattice, including electron microscopy (EM) [27,28]. As fluorescent labelling and imaging advanced, the quantification of immunostained CA relative to fluorescently labelled viral components by microscopy became a method for assessing capsid uncoating kinetics (Table 2) [29].

Assessment of HIV-1 CA mutants has also provided insight into capsid morphology and stability, as well as viral replication. Hyper- and hypostable capsids have allowed the comparison of replication outcomes under these two structural extremes. Amino acid substitutions that result in hypostable capsids, such as K203A, disassembled rapidly post-fusion and failed to complete reverse transcription [17]. Conversely, hyperstable capsids, such as those caused by the mutation E45A, dissociated slowly relative to wildtype capsids and were associated with nuclear import deficiencies [26,39,40]. Intentional stabilization of capsid was also applied to improve its purification. Ultracentrifugation of virions through a sucrose gradient is a common method of capsid isolation. However, a proportion of capsids disassemble during this process, lowering total yield [41]. The incorporation of A14C/E45C, A42C/T54C, and other cysteine residues addressed this limitation by disulfide crosslinking to improve capsid stability [41].

## 3. Capsid Trafficking to the Nucleus and Nuclear Import

### 3.1. Host Factors Involved in Capsid Trafficking

Directed HIV-1 capsid movement involves its binding to cytoskeleton motor proteins and adapters [42,43,44,45,46]. Live cell microscopy using cells infected with fluorescently labelled HIV-1 and expressing fluorescently labelled tubulin identified HIV-1 movement along microtubules, suggesting motor proteins facilitated viral particle trafficking to the nucleus [47]. Fixed cell imaging in this study also demonstrated the close association of GFP-tagged HIV-1 Vpr proteins with stained microtubules, consistent with this hypothesis. Retrograde HIV-1 movement to the interior of the cell was later attributed to the motor protein dynein, as siRNA depletion of dynein heavy chain led to the accumulation of immunostained CA foci at the periphery of cells with decreased nuclear import [48]. The interaction of dynein with the HIV-1 capsid occurred through bicaudal D2 (BICD2), which acts as a dynein adapter for cellular cargoes [43].

Anterograde movement, or trafficking towards the cell periphery, was also observed for labelled HIV-1 PICs by fixed cell imaging, which determined that viral complexes use the kinesin motor protein, KIF5B, to reach the periphery of the nucleus [46,49]. Binding of HIV-1 to KIF5B was later linked to fasciculation and elongation protein zeta-1 (FEZ1), a kinesin-1 adapter protein that binds multiple charged surfaces on the capsid [42,50]. How FEZ1 binds the capsid was resolved by mixing CA labelled with either SpyTag or SpyCatcher components [51]. In close proximity, SpyCatcher-CA became covalently bound to SpyTag-CA, preventing CA molecules from dissociating, allowing the formation of stable pentamers, hexamers, and larger hexamer lattice structures [52].

Motor protein interactions lead to HIV-1 particles that move both towards and away from the cell nucleus, with the depletion of either kinesin or dynein resulting in decreased HIV-1 CA in the nucleus [46,48]. While anterograde movement may seem counterproductive for nuclear entry of the HIV-1 genome, kinesin may allow the redirection of a cargo when dynein encounters physical obstacles [53]. Considering the macromolecular density of the cytoplasm, cargoes that move multi-directionally may increase the chance of reaching the nucleus. Furthermore, the dual forces exerted on the capsid have been proposed to help initiate uncoating by pulling the capsid in opposing directions [46]. The full extent of trafficking can be visualized by fluorescent tagging of PIC components paired with live cell microscopy. These advances in methodology can estimate the trafficking speed and directionality of HIV-1 complexes as well as detect HIV-1 PIC association with host factors in real time.

### 3.2. Host Proteins That Bind to Capsid in the Cytoplasm

Beyond motor proteins, the HIV-1 capsid interacts with other proteins prior to reaching the cell nucleus, such as cyclophilin A (CypA). CypA normally functions as a peptidyl proline isomerase with roles in protein folding, trafficking, and T lymphocyte activation [54]. In the context of HIV-1 infection, CypA binds to an exposed proline residue found on a loop between helices 4 and 5 of HIV-1 CA [55,56,57,58,59]. CypA can promote early HIV-1 replication steps in a cell-type dependent manner [60,61,62]. Despite decades of work, the mechanism of how CypA promotes HIV-1 replication remained unclear until recently, when it was identified to prevent TRIM5α from binding to the capsid surface in primary human CD4^+^ T cells and macrophages [63,64]. In the absence of CypA binding, TRIM5α oligomerized on HIV-1 capsid, leading to capsid destabilization [65,66]. The interplay between CypA and TRIM5α was demonstrated by both virus infectivity assays comparing WT HIV-1 and the P90A CA mutant (which does not bind to CypA) and shRNA depletion of host factors [63]. Fluorescence microscopy supported these data by indicating that TRIM5α proximity to HIV-1 PICs increased with the loss of CypA binding. Biochemical assays with in vitro CA tubular assemblies showed that CypA binding reduced TRIM5α binding [64].

CypA steric hinderance is not exclusive to TRIM5α, as cleavage and polyadenylation specificity factor 6 (CPSF6) binding to the HIV-1 capsid is inhibited by CypA. While CPSF6 predominantly localizes to the nucleus, where it functions as an RNA splicing factor, it was visualized in the perinuclear region of the cytoplasm [67,68]. CPSF6 binds to the HIV-1 capsid through multiple interactions with a hydrophobic pocket formed by helices 3, 4, and 5, in which alteration of key residues, such as N74, abolish binding [24,69,70]. CypA binding to the HIV-1 capsid was shown to decrease the CPSF6–capsid association just outside of the cell nucleus, which was also demonstrated with in vitro CA tubular assemblies and purified CPSF6 and CypA [67]. Capsid and CA assemblies were disrupted upon binding to CPSF6, suggesting that this host factor plays a role in capsid uncoating [67,71]. The competitive interactions of CypA and CPSF6 were quantified by using live cell microscopy, by which fluorescently labelled CPSF6 colocalized with trafficking PICs, presumably through binding to the capsid, in the cell cytoplasm after infection. In HeLa cells and CD4^+^ T cells, CypA expression was localized to the periphery of the cell and excluded in the perinuclear and nuclear regions where CPSF6 is expressed. HIV-1 trafficking in the cytoplasm was altered when CypA binding was disrupted. Fluorescently labelled CypA bound to trafficking HIV-1 capsids was lost during nuclear import [40]. The transition of capsid-bound factors suggests that a capsid handoff between host factor occurs in the cytoplasm, which may include CPSF6 and CypA to prevent premature uncoating. Additional host factors may alter capsid trafficking speed and promote the nuclear import of the viral genome near the nucleus. While we have only discussed CypA, TRIM5α and CPSF6, numerous other cellular factors can bind to the HIV-1 capsid and have been extensively described to promote or inhibit HIV-1 infection (reviewed in [72]).

### 3.3. Capsids at the Nuclear Pore

To infect nondividing cells, HIV-1 must traverse the highly regulated NPC to deliver the viral genome into the nucleus. NPCs consist of dynamic collections of nucleoporins (NUPs) that form a multilayer structure between the cytoplasm and the nucleoplasm. A nuclear localization signal (NLS) is typically required on cellular cargoes for active transport through NPCs [73,74]. While multiple HIV-1 PIC components have a NLS, CA was identified to be the primary determinant for nuclear import by the replacement of murine leukemia virus (MLV) CA with HIV-1 CA [75]. MLV is a gammaretrovirus that is unable to infect nondividing cells. However, MLV with HIV-1 CA was able to infect growth-arrested cells [75]. Since this key work, the field has made strides in understanding the engagement of the HIV-1 capsid with the NPC and delivery of the PIC into the nucleus.

Cytoplasmic NUPs engage the HIV-1 capsid during infection. In live cell imaging experiments, fluorescently labelled PICs switched from active transport into the nucleus to docking at the NPC [76,77]. By producing the virus in cells expressing CypA with a tetrameric dsRed tag, the capsid was indirectly labelled by CypA-dsRed (Table 2). Live cell imaging with HIV-1 labelled with CypA-dsRed and IN-GFP further resolved this distinct docking behavior [40]. Docking was observed as actively transported capsids reached the cytoplasmic side of the nuclear envelope, where PICs remained confined to the nuclear envelope before entering the nucleus [78]. Successful docking remained dependent on capsid, as hyperstable CA mutations, such as E45A, led to erratic capsid docking without stationary docking [40]. To facilitate import and docking, host factors have been shown to be involved, including NUP358 (RANBP2) [79,80,81,82,83].

NUP358 is a large NUP that localizes to the cytoplasmic basket of NPCs. Depletion of NUP358 led to decreased HIV-1 infectivity, indicating a functional role in early replication, specifically at nuclear import [80]. Numerous reports suggested an interaction between NUP358 and the capsid to facilitate docking [40,80,84]. One model proposed that NUP358 shuttles away from the nuclear envelope, using KIF5B to localize with HIV-1 PICs before returning to the nuclear envelope [84]. However, most reports observed HIV-1 PICs with CA at the nuclear envelope with NUP358 [76]. NUP358 contains a C-terminal cyclophilin homology domain that can bind the HIV-1 capsid [79,85,86]. However, the importance of the CypA homology domain is unclear, as some work suggested that this domain is dispensable, with a phenylalanine-glycine (FG)-rich repeat domain sufficient for nuclear import [87]. Since FG repeats facilitate the interaction and delivery of transportins through the NPC, NUP358 FG repeats may support HIV-1 capsid docking and nuclear import in a similar manner.

### 3.4. CPSF6, TNPO3, and Nuclear Import

Transport of the HIV-1 genome into the nucleus is partly dependent on the beta-karyopherin transportin 3 (TNPO3) [88,89]. TNPO3 ferries cargo proteins containing arginine-serine (RS) repeat domains into the nucleus, including CPSF6 [90]. As CPSF6 binds HIV-1 capsid, TNPO3 transports CPSF6 and the bound capsid to the nucleus [91]. Truncation of CPSF6 to disrupt TNPO3 binding or depletion of TNPO3 led to the accumulation of CA in the cytoplasm and a significant decrease in HIV-1 nuclear import products [24,91,92]. Fluorescent tagging of CPSF6 confirmed that TNPO3 is required for CPSF6 localization in the nucleus [91]. TNPO3 depletion led to CPSF6 accumulation in the cytoplasm, which restricted HIV-1 infection [91,93]. Successful nuclear import, however, does not rely solely on CPSF6 and TNPO3. The mutation of N74 or A77 on CA alleviated the CPSF6/TNPO3 requirement for nuclear import, indicating that other factors are involved [24,94].

Once on the nuclear side of the NPC, NUP153 binds the capsid at the CPSF6 binding interface [76,81,82]. Super-resolution microscopy demonstrated the close proximity of NUP153, CA, and CPSF6 in macrophages [95]. Current models suggested that NUP153 may assist in moving the capsid and PIC through the NPC central pore into the nucleus [95]. Since NUP153 and CPSF6 bind the capsid using an FG motif, Bejarano et al. proposed that nuclear CPSF6 competes for capsid binding to release the PIC from NUP153 after nuclear import. Despite studies characterizing NUPs individually, nuclear import remains a complex process with changing HIV-1 dependence based on the NUP composition of the NPC [96]. Other NUPs have been implicated in altering infectivity and nuclear import, such as NUP98 and NUP214, though these factors remain less characterized compared to NUP358 and NUP153 [76,81].

HIV-1 PICs retain some capacity for nuclear import when select NUPs are lost, such as NUP358 [96]. This is likely attributed to multiple NUPs that can interact with CA as well as NPCs dynamically reorganizing in response to NUP depletion, thereby maintaining their functionality. Endpoint infectivity coupled with CA immunostaining identified that NPCs dynamically changed following loss of select NUPs [96]. These changes were reflected in the variable nuclear import of HIV-1 PICs. Thus, elucidating which NUPs are involved in nuclear import remains complex, but is required for understanding how the HIV-1 PIC is delivered into the nucleus.

## 4. HIV-1 Capsid Uncoating and Reverse Transcription Complete in the Nucleus

### 4.1. Limits of HIV-1 CA Detection

Until recently, models of HIV-1 capsid uncoating suggested that it occurs entirely in the cytoplasm. Early isolations of PICs failed to co-purify significant levels of CA, which suggested that CA was rapidly lost after HIV-1 entry [97,98]. Contrary to this model, HIV-1 nuclear entry was shown more recently to be dependent on the capsid [75]. The initial model was likely a result of limitations of the techniques of the time. For example, subcellular fractionation conditions used in those studies likely led to the loss of CA during sample preparation due to the low stability of intact capsid. Studies have since detected CA in the cytoplasm and the nucleus of HIV-infected cells, supporting a role of the capsid in later replication steps. For example, fluorescent labelling of HIV-1 proteins thought to be inside the capsid, such as Vpr and IN, coupled with CA immunostaining measured CA retention and loss over time in situ (Table 2) [29]. However, colocalization of CA with labelled PIC components has often been limited to the cytoplasm.

Cellular assays, such as the cyclosporine A (CsA) washout assay, further supported a model in which HIV-1 CA remains with the PIC in the cytoplasm up to 4 h post-infection (Table 1) [32]. The CsA washout assay requires cells stably expressing TRIM-Cyp, a nonhuman primate fusion of TRIM5α and CypA, that binds to the HIV-1 capsid and restricts infection [12,99]. HIV-1 infection is carried out in the presence of CsA, which disrupts CypA and TRIM-Cyp capsid binding. By removing CsA from the cells at different time points, TRIM-Cyp binding to capsids can restrict infection prior to uncoating. The final readout is endpoint infectivity, allowing uncoating kinetics to be quantified. This method supported a cytoplasmic uncoating model. However, the CsA washout assay has its limitations. During CsA treatment, endogenous CypA binding is lost, possibly leading to alteration of HIV-1 replication. Furthermore, TRIM-Cyp predominantly localizes to the cytoplasm, which limits the ability to address nuclear capsid uncoating. Alternative models have since been proposed in which HIV-1 capsid uncoating occurs at the nuclear envelope or in the nucleus.

Imaging strategies have provided HIV-1 capsid tracking, particularly in the cytoplasm. Direct tagging of HIV-1 CA with fluorescent proteins does not allow the efficient formation of capsids, limiting the detection of CA by immunostaining in fixed cells. While indirect labeling of the capsid with CypA-dsRed was detected in the nucleus with fixed cell imaging, the signal remained relatively weak (Table 2) [36,40]. The lack of substantial nuclear CypA-dsRed with CA further supported a model in which the capsid dissociates in the cytoplasm. Alternatively, fluorescent proteins can be packaged into the HIV-1 capsid during viral production, such as an internal fluorescent protein known as iGFP, in which the GFP gene is genetically inserted into *gag* between cleavage sites (Table 2) [37]. The fluorescent protein is proteolytically cleaved from the Gag polyprotein and is maintained with the capsid until disassembly, as measured by the dissociation of a second fluorescent marker.

### 4.2. Capsids in the Nucleus

As the HIV-1 capsid had been thought to be excluded from the nucleus, with uncoating preceding nuclear import, low levels of CA in the nucleus were viewed as residual protein remaining with the PIC [78,86]. The restriction of CA to the cytoplasm was supported at the time by a lack of detection of intact nuclear capsids and the reported size limitation of 39 nm for the central channel of NPCs [100,101]. However, cryo-electron tomography was used to reconstruct purified nuclear envelopes containing NPCs, which were found to dilate up to 64 nm, theoretically sufficient for the passage of HIV-1 capsids, which can reach 60 nm at the wide end [102]. Additionally, intact capsid lattices were identified within the central NPC pore and in the nucleus by EM. These results suggest that the central NPC pore may dilate enough for intact or partially intact capsids to enter.

Recent use of GFP-CA incorporation into HIV-1 particles allowed CA to be tracked into the nucleus (Table 2) [38]. By inserting GFP between MA and CA in *gag*, the GFP tag was fused to the CA NTD following proteolytic processing. Capsids must contain untagged CA at an optimal ratio with GFP-CA to form mature capsids that lead to infectious viruses and to remain fluorescent. GFP-CA remained associated with PICs until integration, with GFP intensity remaining relatively stable. Importantly, as GFP intensity was not lost over time, this suggested that a fully or nearly intact capsid dissociates shortly before integration [38,103]. Additional studies utilizing encapsulated fluid phase iGFP indicated that the fluorescent signal was retained until after nuclear import, further supporting nuclear capsid uncoating [103]. However, even with an optimized ratio of GFP-CA to untagged CA, infectivity was impaired compared to WT HIV-1. Despite the caveats of GFP-CA, multiple techniques now provide evidence of nuclear capsid entry and uncoating.

Kinetic analysis has also suggested that the HIV-1 capsid is required until after nuclear import by the HIV-1 nuclear import kinetics (NIK) assay [104]. The NIK assay utilizes a chimeric Nup68 fused with an inducible homodimerizing DmrB domain, which functionally blocks NPCs in the presence of the B/B homodimerizer drug. The drug can therefore be added at different timepoints, with infectivity as an indirect measure of nuclear import success. Surprisingly, the inclusion of a capsid inhibitor after nuclear import was sufficient to block infection in multiple cell lines, indicating that the intact capsid is required in the nucleus for infection.

### 4.3. Reverse Transcription in the Nucleus

A defining characteristic of retroviral infection is reverse transcription. While this process can begin within a mature virion, the vast majority of reverse transcription occurs within cells [105]. HIV-1 reverse transcriptase (RT) facilitates the synthesis of ssRNA to dsDNA while degrading the RNA template [106]. RT requires the utilization of host nucleotides, likely while remaining compartmentalized with viral RNA, such that HIV-1 RT remains associated with the RNA genome within an intact capsid [17,107].

Reverse transcription requires access to deoxynucleotides (dNTPs), which must be accessible to RT within the intact capsid. While the capsid was once thought to block the access of metabolites, the discovery of the positively charged hexamer pore provided a mechanism for nucleotide import into the capsid [108]. With each CA monomer contributing a single R18 residue, a positively charged ring is formed for the recruitment of negatively charged dNTPs [108]. The polyanionic metabolite inositol hexakisphophate (IP_6_) and dNTPs were later identified to bind the capsid pore and increase capsid stability [34,108,109]. A recent model for dNTP translocation suggested that the recruitment of dNTPs relies on binding to R18, with K25 facilitating passage through the pore [34]. Molecular dynamic simulations showed binding of two dNTPs in the pore through R18 and K25, which increased translocation. By using the charge-reversed K25N, dNTP recruitment was abolished, further supporting K25 in dNTP binding. dNTP recruitment also showed an increase in capsid stability, potentially by stabilizing the hexamer ring. By quantifying capsid stability with the CA retention assay, K25N HIV-1 was found to retain less CA relative to wildtype virus, confirming the stabilizing roll of dNTP binding (Table 1) [34]. While the capsid structure promotes reverse transcription, the initial loss of capsids is associated with reverse transcription initiation, which highlights the level of interplay between ongoing replication steps [31,32,110,111].

As new evidence of nuclear uncoating has been reported, the spatiotemporal process of reverse transcription has also shifted. Under the older cytoplasmic uncoating models, reverse transcription was presumed to progress in the cytoplasm prior to nuclear import. However, reverse transcription kinetics have been found to lag behind nuclear import in the NIK assay [104]. By blocking reverse transcription with an RT inhibitor at various time points, nuclear import was found to occur prior to reverse transcription completion. This suggests that reverse transcription remains an ongoing process, even after nuclear import.

Several studies have utilized fluorescence microscopy and biochemical approaches to capture reverse transcription localization. Nuclear reverse transcription progression was visualized by staining PICs with the 5-ethynyl-2′-deoxyuridine (EdU) nucleotide analog [112,113]. As reverse transcription progressed, EdU was incorporated into nascent viral DNA. EdU-positive PICs were detected as an indicator of partial or complete reverse transcription products. In the presence of RT inhibitors, the nuclear EdU signal decreased, suggesting that reverse transcription occurred in the nucleus [114]. An alternative approach to visualize reverse-transcribed DNA with capsid used the ANCHOR system, which relies on incorporation of a modified bacterial ParS sequence (ANCH) into the HIV-1 genome [115]. The ParS binding partner, ParB, can be fused to GFP (OR-GFP) and expressed in cells. Thus, with this HIV-1 labelling approach, cellular OR-GFP bound to the ANCH sequence, allowing HIV-1 DNA detection with labelled IN and CA in the nucleus.

### 4.4. Biphasic Uncoating

Early definitions of uncoating often described the event as a one-step loss of CA monomers [97,98]. However, uncoating is now viewed as a set of complex events associated with reverse transcription and nuclear import. It is no longer sufficient to describe uncoating as a one-step loss of CA. Rather, an initial loss of CA is detectable that precedes a larger disassembly event. This has been termed biphasic uncoating due to the two-step behavior. Biphasic uncoating was first reported using 5-ethynyl uridine (EU) labelled virions (Table 2) [35]. Incorporation of the ribose analog into HIV-1 RNA allows ligation of fluorescent azide dyes, which are excluded from the capsid. This exclusion of RNA staining was found to be size-selective with capsids, allowing dye labelling but not the entry of the protein. Importantly, EU staining was not representative of complete disassembly, as EU labelled HIV-1 RNA was found to be resistant to RNase H degradation. EU staining was also observed with the hyperstable E45A capsids, suggesting that the initial capsid opening is separate from complete disassembly. Biphasic uncoating was later reproduced with single molecular fluorescence imaging of Gag-iGFP (Table 2) [33]. This in vitro method provides live analysis of cell-free capsids with encapsulated fluid phase iGFP. Loss of the internal marker denoted capsid opening, while CypA-dsRed labelled the larger CA lattice until full disassembly. Distinct disassembly events consisting of an initial loss of CA followed by larger lattice loss were observed, supporting a model of biphasic change over time in capsids.

Multistep changes in capsids have also been reported through biochemical analysis by atomic force microscopy (AFM) [116]. Using AFM, capsid stability can be probed by in vitro stiffness measurements of the HIV-1 capsid surface, in which a change in capsid stiffness corresponded with initiation of reverse transcription as measured by qPCR (Table 1) [31]. Peak stiffness decreased over several hours and correlated with capsid opening and disassembly when imaged by EM. Inhibition of reverse transcription ablated capsid opening, linking morphological changes to reverse transcription progression as previously reported. Similarly, the E45A capsid mutant underwent similar stiffness changes that corresponded with capsid opening, though complete disassembly was delayed. The alteration of the capsid structure has been suggested as necessary for nuclear import during nuclear docking [40]. Recent detection of nearly complete capsids in the nucleus supports this model, although timing of partial CA loss has not been fully resolved between nuclear entry and post-entry steps [38,102,117,118].

## 5. Nuclear CPSF6, Capsid, and Integration

A hallmark of retroviral infection is the deliberate, stable integration of the reverse-transcribed viral genome into the host cell genome that establishes persistent infection of the cell and its progeny [119]. Once translocated into the nucleus, the PIC is targeted to the chromosomal genome, where HIV-1 IN performs a DNA strand transfer reaction to facilitate integration [120]. Hydrolysis of chromosomal DNA is then facilitated by the 3′ processed ends of the viral DNA, covalently linking the proviral DNA. DNA repair machinery facilitates the repair of the remaining 3′ gaps, thereby completing integration [121,122].

HIV-1 preferentially integrates into transcriptionally active sites [123,124]. While HIV-1 IN is sufficient to perform the integration reaction, it requires assistance from host factors to target particular regions in the host genome. Nuclear CPSF6 was shown to bind to CA, allowing PICs to penetrate the interior of the nucleus and target active transcription units [125,126,127]. PIC localization was linked to transcriptionally active chromatin with speckle-associated genomic domains (SPADS) [128]. Nuclear PICs were observed as being colocalized with fluorescently tagged CPSF6, appearing as defined CPSF6 puncta [38,117,128]. Disruption of capsid structure by PF74 treatment led to a loss of CPSF6 puncta, indicating that the capsid lattice is required in this interaction. By immunostaining for nuclear speckle-associated proteins, colocalization of PICs with nuclear speckles prior to integration was visualized [128]. Loss of CPSF6 binding through the N74D or A77V CA mutations led to a failure to localize with SPADs and CPSF6, highlighting the importance of this interaction in integration targeting. The N74D and A77V CA mutations resulted in PICs remaining near the nuclear envelope, which corresponded to integration into areas of lower transcription activity [40,83,86,107,127].

## 6. Conclusions

The HIV-1 capsid facilitates numerous viral–host interactions within the cell that determine replication success. Decades of advancement in the tools available to detect CA, measure uncoating, quantify capsid stability, and determine interactions with host factors have increased the understanding of the HIV-1 capsid dramatically. With these advances, models of early HIV-1 replication have recently shifted. The most recent research defined the completion of capsid uncoating and reverse transcription in the nucleus (Figure 1). As the field moves forward to study nuclear capsids and reverse transcription, preexisting and novel methods will remain critical for fully understanding HIV-1 infection.

## Figures and Tables

**Figure 1 viruses-13-02237-f001:**
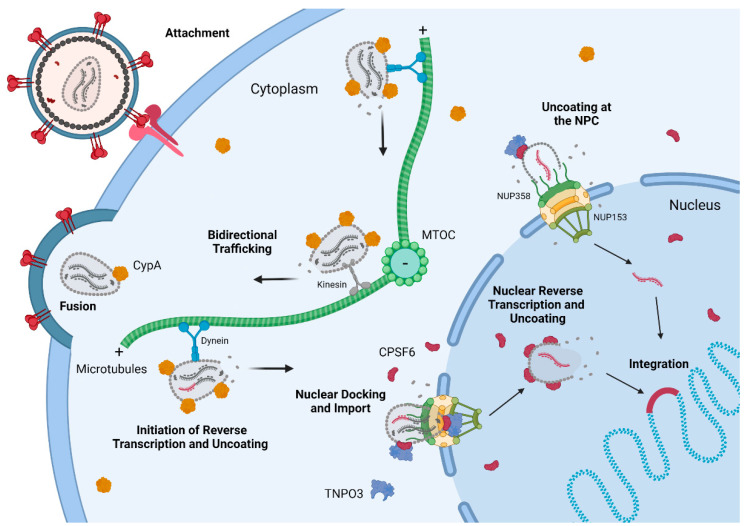
Overview of early HIV-1 replication. HIV-1 attaches to the CD4 receptor and chemokine co-receptor before fusing with the cell membrane, releasing the capsid into the cytoplasm where host factors bind the capsid surface and capsid uncoating may initiate. Dynein and kinesin facilitate bidirectional movement of the HIV-1 capsid on microtubules. Near the nucleus, the capsid is bound by CPSF6 and TNPO3. Engagement with the NPC allows capsid docking and eventual nuclear import. In the nucleus, reverse transcription and capsid uncoating are completed near the site of integration. CypA, cyclophilin A; MTOC, microtubule organizing center; TNPO3, transportin 3; CPSF6, cleavage and polyadenylation factor 6; NPC, nuclear pore complex. Created using BioRender.

**Table 1 viruses-13-02237-t001:** Biochemical, in vitro imaging, and infectivity assays to assess HIV-1 capsid uncoating/integrity.

Category	Assay (Lab)	Principle	Pros	Cons
**Biochemical**	In vitro capsid stability assay (Aiken) [26]	Isolated virions are ultracentrifuged over a sucrose gradient to separate intact capsids from disassembled capsids; western blot of fractions provides a bulk readout of capsid populations	Direct measurement of intact capsids Not technically challenging	Does not determine infectivity Some intact capsids are lost during the sample prep and ultracentrifugation Population measurement that does not allow visualization of individual particles or their kinetics
	Fate of the capsid (Sodroski/Diaz-Griffero) [30]	Western blot of pelletable vs. soluble CA isolated from infected cells	Theoretically not technically challenging Cellular assay that can be performed in any cells	Does not differentiate between intact capsids or aggregated CA Does not determine infectivity Population measurement that does not allow visualization of individual particles or their kinetics
	Atomic force microscopy of capsids (Rousso) [31]	Pressures of individual capsids are measured by atomic force microscopy	Analysis of individual capsids Quantifies structural changes over time	Specific equipment required Does not determine infectivity
**Infectivity**	CsA washout assay (Hope) [32]	Cells that express TRIMCyp are infected in the presence of cyclosporine A (CsA), which is washed out at different time points prior to measurement of infectivity	Not technically challenging and no special equipment needed Measures infectivity	Requires cells expressing TRIMCyp Population measurement that does not allow visualization of individual particles May not accurately reflect uncoating of CsA dependent CA mutants Limited to cytoplasmic capsids
**In vitro imaging**	Single-molecular fluorescence imaging of Gag-iGFP (Böcking) [33]	HIV-1 containing Gag-iGFP (with or without CypA-dsRed) is imaged for retention of iGFP by total internal reflection fluorescence (TIRF) microscopy	Provides kinetics of many individual capsids Can evaluate contribution of cell factors and drugs on capsid integrity	Requires confocal microscopy Does not determine infectivity Quantifies loss of fluorescence
	CA retention assay (Ambrose) [34]	Fixation and permeabilization of HIV-1 containing a fluorescent capsid marker, followed by CA staining and TIRF imaging for CA retention	Provides kinetics of many individual capsids Can evaluate contribution of cell factors and drugs on capsid integrity	Requires confocal microscopy Does not determine infectivity Quantifies loss of fluorescence

**Table 2 viruses-13-02237-t002:** Fixed and live cell microscopy assays to assess HIV-1 capsid uncoating/integrity.

Category	Assay (Lab)	Principle	Pros	Cons
**Fixed Cell Imaging**	In Situ uncoating assay (Hope) [29]	Cells are infected with HIV-1, followed by fixation and staining of CA protein	Cellular assay that can be performed in any cells Visualization of actual CA/capsid	Staining may be variable depending upon antibody used Does not allow visualization of individual particle kinetics Quantifies loss of fluorescence
	EU staining assay (Ambrose) [35]	Cells are infected with HIV-1 produced in the presence of 5-ethynyl uridine (EU) and a second marker, followed by fixation and staining of EU	Cellular assay that can be performed in any cells Measures gain of fluorescence signal	Does not allow visualization of individual particle kinetics
**Live Cell Imaging**	CypA-DsRed live cell imaging (Melikian) [36]	Cells are infected with HIV-1 made in the presence of CypA-DsRed and a second marker, followed by imaging of loss of CypA-DsRed signal	Cellular assay that can be performed in any cells Provides kinetics of many individual capsids	Quantifies loss of fluorescence Does not reflect uncoating of CypA independent CA mutants Limits CA tracking to the cytoplasm
	Gag-iGFP live cell imaging assay (Hope) [37]	Cells are infected with HIV-1 containing Gag-internal GFP (Gag-iGFP) and second marker, followed by imaging of loss of GFP signal	Cellular assay that can be performed in any cells Provides kinetics of many individual capsids	Quantifies loss of fluorescence
	GFP-CA live cell imaging assay (Pathak) [38]	HIV-1 is produced by phenotypic mixing of WT CA and GFP-CA and second marker, followed by imaging of loss of GFP-CA signal	Cellular assay that can be performed in any cells Provides kinetics of many individual capsids	Virus has decreased infectivity Not all CA is labelled with GFP Quantifies loss of fluorescence

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
