# Peer review of "Disassembling the Nature of Capsid: Biochemical, Genetic, and Imaging Approaches to Assess HIV-1 Capsid Functions"

_viruses, 2021, doi:10.3390/v13112237_

Round 1
Reviewer 1 Report
The manuscript entitled “Disassembling the Nature of Capsid: Biochemical, Genetic, and Imaging Approaches to Assess HIV-1 Capsid Functions" authored by Zachary Ingram and colleagues is well written and suitable for publication.
The review is a comprehensive summary of the present advances achived in the field.
Author Response
We thank the reviewer for their assessment.
Reviewer 2 Report
The review written by Ingram et al. with the title “Disassembling the Nature of Capsid: Biochemical, Genetic, and Imaging Approaches to Assess HIV-1 Capsid Functions” is a very well written and comprehensive review paper. Authors talk about HIV-1 Capsid, an important structure in HIV-1 which contribute to HIV-1 pathogenesis and every step of infection. The authors reviewed the information available in literature about the capsid function and summarize the methods to study Capsid structure and function by different methods. The information contributes to provide better therapeutics against HIV-1. One important contribution they missed and should include in their paper is
Campbell EM, Hope TJ. HIV-1 capsid: the multifaceted key player in HIV-1 infection. Nat Rev Microbiol. 2015;13(8):471-483. doi:10.1038/nrmicro3503.
More than 50% of all cited references in the review are not recent and more than 5 years old. Recent literature will be a better option in providing latest information in this field.
Author Response
We included the Campbell and Hope 2015 reference as suggested.
As this was a comprehensive review of assays used to study HIV-1 capsid, we cited many relevant, historical references. The bibliography includes 51 references from the past 5 years. Another 33 references are from the past 10 years (66%).
Spellcheck was performed on the manuscript.